# A protocol for the economic evaluation of the smoking Cessation in Pregnancy Incentives Trial III (CPIT III)

Nicola McMeekin ,[1] Lesley Sinclair,[2] Linda Bauld,[2] David Michael Tappin,[3] Alex Mitchell,[4] Kathleen Anne Boyd[1]

[1]Health Economics and Health Technology Assessment, Institute of Health and Wellbeing, University of Glasgow, Glasgow, UK
[2]Usher Institute of Population Health Sciences and Informatics, The University of Edinburgh, Edinburgh, UK
[3]Scottish Cot Death Trust, West Glasgow Ambulatory Care Hospital, University of Glasgow, Glasgow, UK
[4]York Trials Unit, Department of Health Sciences, University of York, York, UK

**Correspondence to**
Ms Nicola McMeekin;
nicola.mcmeekin@glasgow.ac.uk

## ABSTRACT

**Introduction** Smoking results in an average 10-year loss of life, but smokers who permanently quit before age 40 can expect a near normal lifespan. Pregnancy poses a good opportunity to help women to stop; around 80% of women in the UK have a baby, most of whom are less than 40 years of age. Smoking prevalence during pregnancy is high: 17%–23% in the UK. Smoking during pregnancy causes low birth weight and increases the risk of premature birth. After birth, passive smoking is linked to sudden infant death syndrome, respiratory diseases and increased likelihood of taking up smoking. These risks impact the long-term health of the child with associated increase in health costs. Emerging evidence suggests that offering financial incentives to pregnant women to quit is highly cost effective.

This protocol describes the economic evaluation of a multi-centre randomised controlled trial (Cessation in Pregnancy Incentives Trial III, CPIT III) designed to establish whether offering financial incentives, in addition to usual care, is effective and cost effective in helping pregnant women to quit.

**Methods and analysis** The economic evaluation will identify, measure and value resource use and outcomes from CPIT III, comparing participants randomised to either usual care or usual care plus up to £400 financial incentives. Within-trial and long-term analyses will be conducted from a National Health Service and Personal Social Services perspective; the outcome for both analyses will be quality adjusted life-years measured using EQ-5D-5L. Patient level data collected during the trial will be used for the within-trial analysis, with an additional outcome of cotinine validated quit rates at 34–38 weeks gestation and 6 months postpartum. The long-term model will be informed by data from the trial and published literature.

Ethics and dissemination

**Trial registration number** ISRCTN15236311; Pre-results (https://doi.org/10.1186/ISRCTN15236311).

## Strengths and limitations of this study

► Prospectively designed economic evaluation of a phase III randomised controlled trial with sites across the UK.
► Preference-based (utility) outcome measures at late-stage pregnancy and 6 months postpartum to enable decision-making.
► Six months postpartum follow-up is the longest we are aware of, making this a novel study.
► Lifetime extrapolation includes costs and outcomes for both mother and infant.
► Challenges relate to different smoking cessation service delivery at each trial site.

## INTRODUCTION
### Problem

Smoking is the principal preventable cause of cancer, with 64 000 new cases annually in the UK.[1] Levels of smoking reflect levels of inequality; in 2016, in England, 7.9% of adults smoked in the least deprived areas compared with 27.2% in the most deprived areas.[2] Lifelong smokers lose 10 years of life; however, smokers who quit before 40 years of age can expect a near to normal lifespan.[3] Levels of smoking during pregnancy are high; in 2010, 12% of mothers in England smoked throughout their pregnancy; this figure was 13% and 15% in Scotland and Northern Ireland, respectively.[4] Smoking during pregnancy is a recognised cause of low birth weight and increases the risk of premature birth by 27%.[5] In addition to complications during pregnancy, after birth the consequences of continued smoking on the child are substantial: passive smoking is linked to sudden infant death syndrome, lower respiratory diseases, asthma and impaired lung function.[5] Research shows that children in a house where parents or siblings smoke are 90% more likely to smoke themselves.[6] Around 80% of women in the UK have a baby,[7] most of whom are less than 40 years of age. Pregnancy is a good opportunity to help women quit smoking, and also decreases the likelihood of the baby becoming a smoker in later life.[6] Helping pregnant women to quit smoking can potentially help tackle inequalities and lift families out of poverty.[8] However, few of about 130 000 pregnant smokers in the UK quit. In the UK, all pregnant women are

offered National Health Service (NHS) smoking cessation services, with free nicotine replacement therapy (NRT); however, only 10% use these services and as few as 3% report abstinence at 4 weeks after quitting.[9]

### Evidence

Financial incentives are becoming a popular approach to changing behaviour that will lead to healthier lifestyles.[10] A recent Cochrane review assessed incentives for smoking cessation and identified nine studies reporting results for pregnant women with a combined relative risk of 2.38 (95% CI: 1.54 to 3.69) in favour of incentives. Eight studies were based in the USA, and one in the UK. The UK study was a single site study (Cessation in Pregnancy Incentives Trial II, CPIT II), which found that offering financial incentives helped pregnant women in Glasgow, Scotland, to quit.[11] CPIT II estimated a cost per quality adjusted life-year (QALY) of less than £500, which is considered to be highly cost effective.[12]

### Aim

CPIT III evaluates whether the favourable findings from CPIT II can be replicated in further UK sites, and includes an additional follow-up at 6 months postpartum. The CPIT III main study protocol is published elsewhere.[13] The aim of the economic evaluation described here is to explore the cost effectiveness of including financial incentives alongside usual care to increase the quit rate of pregnant women who smoke.

## METHODS AND ANALYSIS

### The main randomised controlled trial (CPIT III)

CPIT III is a multicentre randomised controlled trial, which aims to assess the effectiveness and cost effectiveness of adding financial incentives to usual care to increase the smoking cessation rate among pregnant women. The primary outcome is smoking at 34–38 weeks gestation, self-reported, with those reporting as quit undertaking urine or saliva confirmatory biochemical testing. The secondary outcomes comprise: engaging with smoking cessation services and setting a quit date; smoking cessation at 4 weeks after quit date and 6 months postpartum

(with those reporting as quit confirmed by biochemical testing); continuous abstinence from late pregnancy to 6 months postpartum; birth weight; cost effectiveness and process evaluation.

Participants are pregnant women who self-report as smokers, are 16 years or older, less than 24 weeks pregnant and English speaking. The target recruitment number was 940, to give 90% power and 5% significance to show a doubling of quit rate from 7% in the control arm to 14% with financial incentives, allowing 15% loss to follow-up. The estimate of 7% in the control arm was derived from CPIT II (8.6% abstinent)[11] and two other recent large trials of pregnancy cessation interventions, conducted in regions where CPIT III will recruit: abstinent 6.4%[14] and 7.6% .[15] The average control group abstinence rate was 7.5% at late pregnancy. The estimate of 14% in the intervention arm is based on both the cessation rate in CPIT II (22.5%) plus a reflection of what is considered clinically important. A gain of 7%–14% would be considered clinically important and is comparable to pharmaceutical aids in non-pregnant smokers.[16] Assuming participants with missing smoking status are smoking as per the Russell Standard,[17] the attrition rate would be 0%, and as a result 940 participants would give 94% power to detect a doubling of the quit rate from 7% to 14%.

944 participants were recruited from seven sites in England, Scotland and Northern Ireland between February 2018 and April 2020. Pregnant smokers identified at routine maternity booking appointments who were referred to smoking cessation services were assessed for eligibility. The trial was introduced to eligible women at first routine contact with smoking cessation services. Those interested in taking part were sent a participant information sheet and contacted by telephone after at least 5 days to obtain formal consent. Those consenting were randomised to either receive financial incentives plus usual smoking cessation support or usual smoking cessation support only. The incentives are presented in table 1.

Both groups will be offered local smoking cessation support and a research participation voucher. This shopping voucher for £50 is issued if data is provided for

| Table 1 | Intervention and research participation incentives | |
|---|---|---|
| **Time point** | **Intervention** | **Control** |
| Initial local smoking cessation services meeting and setting quit date | £50 voucher | N/A |
| Verified quit at 4 weeks after quit date | £50 voucher | N/A |
| Verified quit at 12 weeks after quit date | £100 voucher | N/A |
| Verified quit at 34–38 weeks gestation and provision of urine/saliva sample if quit | £200 voucher | N/A |
| Providing data for primary outcome | £50 voucher for research participation | £50 voucher for research participation |
| Providing data for secondary outcome (quit 6 month postpartum) | £25 voucher for research participation | £25 voucher for research participation |

the primary outcome in late pregnancy and a further £25 shopping voucher issued if data is provided for the secondary outcome at 6 months postpartum. Participants can still receive the £25 shopping voucher if they provide data for the secondary outcome but do not provide data for the primary outcome.

Participants in the intervention arm will also receive up to £400 in shopping vouchers, £50 for attending an initial appointment with local smoking cessation services and setting a quit date, £50 if verified quit at 4 weeks after quit date, £100 if verified quit at 12 weeks after quit date and £200 if verified quit and provide a saliva/urine sample at 34–38 weeks gestation (primary outcome point). All incentive payments, apart from the first, require quit to be verified by carbon monoxide (CO) breath test. Cotinine/anabasine will also be used to confirm non-smoking status at the primary outcome and 6 months postpartum. Participants in the incentives group not attending smoking cessation services can still receive the final £200 shopping voucher if they are CO verified quit at the primary endpoint of 34–38 weeks gestation and provide a saliva/urine sample.

Further details on the CPIT III are detailed in the main study protocol paper, which is available elsewhere.[13]

### The economic evaluation

The economic evaluation will be conducted from the perspective of the UK's NHS and Personal Social Services (PSS), including only resources funded by the NHS, and include costs for the price year 2020.

Two analyses will be carried out: a within-trial analysis and a long-term lifetime analysis. The within-trial results will be presented as an incremental cost per quitter and an incremental cost per QALY gained. The lifetime model results will be reported as incremental cost per QALY gained. For both analyses, results will be assessed against the current National Institute for Health and Care Excellence (NICE) threshold of £20 000–£30 000[18] per QALY gained. The economic evaluation will adhere to best practice and guidelines.[19 20]

An overview of this analysis is illustrated in figure 1. The participants (pregnant smokers) are randomised to either

the intervention or control arm. The primary outcome of the trial is quit at 34–38 weeks gestation and both this trial primary outcome and the economic outcome of QALYs will be used in the cost-effectiveness analysis.

The analysis will be conducted using the statistical analysis package STATA V.16 (StataCorp LLC, 2019, Stata Statistical Software: Release 16, College Station, Texas, USA).

### Within-trial analysis

Patient level data on resource use and outcomes will be routinely collected during the trial and used in the economic evaluation. The time horizon for the within-trial analysis is that of the trial, from time of maternity booking (less than 24 weeks gestation) to late pregnancy (34–38 weeks gestation) and 6 months postpartum. No discounting of costs and outcomes is required as this time horizon is less than 1 year. The analysis will be undertaken based on an intention to treat approach, and participants lost to follow-up will be assumed to be smokers.

### Resource use

There are four main resource use groups of relevance informing the within-trial analysis: smoking cessation support (professional time), NRT, intervention costs (financial incentives) and neonatal care. Methods of data collection for these resource use categories are described in the following section.

Service delivery of smoking cessation support and prescribing of NRT is likely to vary between trial sites. Health economists will work with each site to establish the format of the service delivered at that site, and set up and maintain resource use data collection procedures. To date, methods used for collecting resources include routinely collected NHS data and bespoke Excel spreadsheets. Resource data for the cessation services will typically be around 12 weeks' worth of NRT and cessation support from first contact with cessation services to last contact.

The trial data management system will record when shopping vouchers are sent to participants and a standard postal cost will be assigned to each voucher posted. Vouchers are sent at first consultation, 4 weeks and 12 weeks after quit, late pregnancy (34–38 weeks) and 6 months postpartum (see table 1). Resource use in terms of shopping vouchers will be recorded at each follow-up point, up to and including 6 months postpartum.

The child's birth weight and date of delivery will be collected postpartum via the trial data management system, enabling preterm status to be derived. Assumptions will be made about the levels of neonatal care required based on this birth weight and prematurity/preterm status.

### Unit costs

Unit costs will be obtained from routine sources:
▶ British National Formulary—NRT.[21]

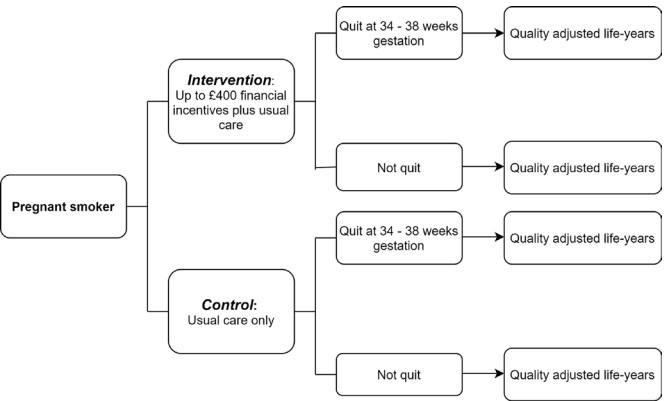

**Figure 1** Decision tree, including intervention and outcomes.

- ► PSS Research Unit—smoking cessation advisor/pharmacist.[22]
- ► Trial data management system—shopping vouchers/postage.
- ► NHS reference costs, Information Services Division NHS Scotland, and the literature—neonatal costs.

Costs will be for the price year 2020, expressed in pounds sterling (£) and inflated if necessary using the Health Services (HS) index.[23] For publication in journals, final costs may also be presented in US dollars, converted using the exchange rate, so that the results will be appropriate for an international audience.

Unit cost information will be combined with the resource use data collected in the trial to estimate the total cost per participant in each trial arm. These participant total costs will be aggregated to estimate the total cost of each trial arm and subsequently the average cost per participant for each intervention.

Regression analysis will be conducted to explore the effect that baseline variables have on the cost of each intervention, such as EQ-5D-5L score, site and gestational age at booking. Variables included will be in line with the statistical analysis plan and main analysis.

### Outcomes

There will be two economic outcomes: the within trial quit rate and QALY.

The primary outcome of the trial is smoking status at 34–38 weeks gestation. This indicator will inform the incremental cost per quitter analysis. As this is an intermediate outcome and not a measure of health, we are also conducting a within trial incremental cost per QALY analysis.

The QALY is a combination of quality of life and quantity of life. Quality of life will be measured using the EQ-5D-5L questionnaire,[24] which measures five dimensions: mobility, self-care, usual activities, pain/discomfort and anxiety/depression, with five levels: no problems, slight problems, moderate problems, severe problems and extreme problems. The EQ-5D-5L will be completed by participants at three time points: baseline, late pregnancy (34–38 weeks gestation) and 6 months postpartum.

The EQ-5D-5L is converted to utilities using value sets giving a utility value of between −0.59 and 1. A utility value of 1 represents full health, 0 represents death and between -0.59 and 0 represents a state worse than death (where members of the general public were asked to value health states using time trade off, one option was believed to be worse than death, ie, bowel and bladder incontinence). We will map the EQ-5D-5L results to the EQ-5D-3L UK value set[25] in accordance to the recommendation in the current position statement from NICE.[26 27]

The quantity of life element of the QALY will be the length of time the participant remains in the trial, using standard QALY calculation applying an area-under-the-curve approach.[28] For example, for a participant recruited at 20 weeks gestation who has primary outcome data collection at 38 weeks, the calculation for a QALY would be as follows:

(Utility at 38 weeks gestation−utility at 20 weeks gestation)×$\underline{38}$−20/52.

When combining the utility score generated from the EQ-5D-5L with quantity of life to give a QALY, standard area-under-the-curve methods[29] will be employed, weighting utilities by the duration of each time interval. Any change in utility between time points will be treated as linear.

A regression analysis will be carried out in line with the main analysis and statistical analysis plan exploring the effect that baseline variables have on the outcomes, such as site, gestational age at booking and in particular baseline utilities.[28]

### Analysis of costs and effects

Missing health economic data will be dealt with in line with best practice guidance[30]: a descriptive analysis of the missing data will be carried out and assumptions will be made to ascertain the type of missingness; following this, an appropriate method for handling missing data will be chosen. If data is missing at random, we will use multiple imputation using chained equations. Finally, a sensitivity analysis for complete cases will be conducted and further sensitivity analyses will be carried out around assumptions as appropriate, for example, using best/worst case assumption scenarios and exploring unexpected issues arising such as COVID-19 and not being able to take CO monitoring readings. Any participants with missing outcome data will be assumed to be smokers.

The cost-effectiveness analysis will bring together the estimates of cost and effect as described above, by estimating an incremental cost effectiveness ratio (ICER) of cost per quitter and per QALY. The difference between mean costs will be divided by the difference in mean QALYs and quit rates to calculate the ICER; the formula for this is presented below. The control arm is denoted by 'A' and intervention (financial incentives) arm by 'B'.

$$ICER = \frac{C_B - C_A}{E_B - E_A}$$

The costs, QALYs and quit rates per participant in each trial arm (ie, the average quit rate per arm) will be presented as means and SD. Differences in means between trial arms will be presented with 95% CIs using generalised linear models with appropriate link and family functions.

### Uncertainty

The effect of uncertainty on the results will be explored using non-parametric bootstrapping techniques and the resulting samples will be plotted on a cost-effectiveness plane in order to graphically represent uncertainty.[31] Cost-effectiveness acceptability curves will then be used to present uncertainty in terms of willingness to pay under various monetary thresholds.[32] Ninety-five per cent CIs for the ICER will be presented using the results of the bootstrapping. If appropriate, 95% CIs for the ICER will

be estimated using Fieller's theorem, a technique that includes any correlation between cost and outcome.[33]

Any subgroup analyses deemed appropriate in the statistical analysis plan will also be considered for the economic analysis. This will include an analysis using self-reported smoker/quitter extracted from case notes, where the trial team have not been able to establish smoking status via direct contact with participants.

Two sensitivity analyses will be run: using miscarriage data from the trial and exploring the effects of gaming (where the participant self-reports as a quitter but is still smoking) in the trial.

### Long-term model

The time horizon for the long-term economic model is lifetime. Costs and effects will be discounted at 3.5% annually in line with recommendations from NICE.[20]

The within trial analysis only considers the costs and effects of the pregnant woman and neonatal costs. The long-term model will also include the costs and effects related to the child. The rationale for this is that children of smokers are more likely to smoke themselves and will also be exposed to second-hand smoke, increasing the likelihood of smoking-related diseases.[6]

### Model structure

The analysis will use and adapt a published probabilistic decision analytic Markov model, the Economics of Smoking in Pregnancy (ESIP) model.[34] The model will comprise two sections: one to capture the lifetime costs and outcomes for the mother, and one to capture the lifetime costs and outcomes of the child.[20] The ESIP model was the first model specifically developed to include future potential cost savings and improvement in health for both mother and child. A short-term cost-effectiveness analysis can be misleading as the benefits of stopping smoking are not captured in a short-time horizon, which is why it is important to incorporate future benefits to assess the full impact of offering financial incentives to pregnant smokers.

The mother's portion of the model will include: the costs of smoking cessation services, financial incentives, NRT use and smoking-related diseases (coronary heart disease, chronic obstructive pulmonary disease (COPD), lung cancer and stroke that account for 75% of smoking related deaths), plus data on relapse rates and transition to smoking-related diseases if the woman relapses. The child's part of the model will include: costs associated at birth for children with smoking mothers if appropriate (premature birth and underweight new-borns), outcomes related to exposure to second-hand smoke in the home (ie, asthma and longer term smoking-related diseases), smoking uptake rates for children of smokers and associated smoking-related diseases.

The trial quit rate at 6 months after birth will be used to predict smoking relapse postpartum. The final utility estimates for patient level data from the postpartum follow-up will also be used as a point for extrapolation

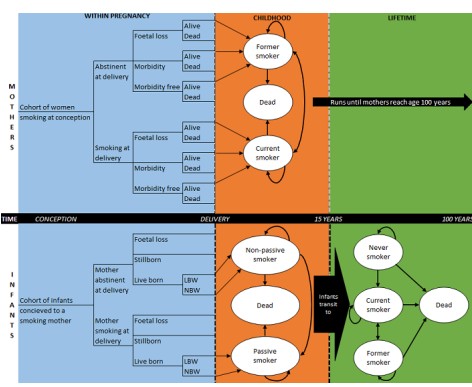

**Figure 2** Model structure (Jones et al[35]). LBW; low birth weight, NBW; normal birth weight.

for lifetime analysis (rather than assuming background population utility values).

Figure 2 depicts the likely structure of the long-term model.[35] The model is split between mother and infant; the first 'within pregnancy' section on the left illustrates within pregnancy outcomes and related costs, the middle 'childhood' section depicts the infant's childhood from birth to 15 years old and the last section 'lifetime' on the right depicts lifetime for both mother and infant.

Resource use and outcomes collected during the trial will be extrapolated and supplemented with published costs from the literature.

### Analysis

The lifetime analysis results will be presented as cost per QALY gained for both the mother and child, separately and combined.

A probabilistic sensitivity analysis will be included and appropriate sensitivity analyses will explore underlying assumptions used in the model, including:
► Re-analysis to account for gaming identified.
► Self-reported outcomes.
► Varying amount of incentive.
► Applying alternative discount rates.
► Varying relapse rates.

### Patient and public involvement

No patient involved.

## ETHICS AND DISSEMINATION

Ethics approval was received from NHS West of Scotland Research Ethics Committee on 15 August 2017. Results of the main trial and economic evaluation will be disseminated through peer-reviewed publications and presentations.

**Acknowledgements** Acknowledgements to the wider CPIT team: Margaret McFadden, Helen Tilbrook, Ada Keding, Judith Watson, Frank Kee, David Torgerson, Catherine Hewitt, Jennifer McKell, Pat Hoddinott, Fiona M Harris, Isabelle Uny and Michael Ussher.

**Authors' contributions** NMM and KAB designed the economic evaluation protocol and wrote the paper. LS, LB, DMT and AM contributed to revising the content of the paper and contributed to editing.

**Funding** This work was supported by Cancer research UK (C48006/A20863), Chief Scientist Office Scottish Government (HIPS/16/1), Health and Social Care Services Northern Ireland (COM/5352/17), Chest Heart & Stroke Northern Ireland (2019_09), Lullaby Trust (272), Scottish Cot Death Trust (no reference available) and Public Health Agency NI (no reference available).

**Competing interests** None declared.

**Patient consent for publication** Not required.

**Provenance and peer review** Not commissioned; externally peer reviewed.

**ORCID iD**
Nicola McMeekin http://orcid.org/0000-0003-2918-8820

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
