## [Reviewer comments · BMJ Open]

ARTICLE DETAILS

TITLE (PROVISIONAL)	A protocol for the economic evaluation of the Smoking Cessation in Pregnancy Incentives Trial (CPIT III)
AUTHORS	McMeekin, Nicola; Sinclair, Lesley; Bauld, Linda; Tappin, David; Mitchell, Alex; Boyd, Kathleen

VERSION 1 – REVIEW

REVIEWER	Mette Rasmussen Bispebjerg and Frederiksberg Hospital, Denmark
REVIEW RETURNED	22-Apr-2020

GENERAL COMMENTS	Dear authors, In general: 1) This is an important research topic evaluating the effects of financial incentives and smoking cessation in a very relevant and vulnerable subgroup of smokers not to mention their unborn children in England, Scotland, and Northern Ireland, using data from a Randomized Controlled Trial (CPIT III). I look forward to reading the results. 2) At first, I was a bit confused about the connection to the CPIT III RCT. I would be nice with a sentence or two in the beginning of the manuscript to explain the connection. I do realize the RCT has been published in a separate protocol, but the most important information regarding the study, including the dates for inclusion and patient consent, should be repeated here. Introduction: 3) Page 5, lines 22-30: The use of references concerning the prevalence of smokers among pregnant women and for the consequences of smoking during pregnancy is sparse. 4) Page 5, line 45ff: In the evidence section you state that "a recent Cochrane review which identified 33 trials (over 21,600 participants) found that nine trials following 2,273 pregnant smokers confirmed the efficacy of financial incentives in this group (Notley)." Only a small part of the studies in this review does, however, confirm the efficacy of financial incentives. I suggest that you add your rationale for choosing to do this study, in spite of the limited evidence. I don't understand what "(Notley)" in the end of the sentence means, I assume it could be the reference to the Cochrane review? 5) Page 6, line 19ff: In line 19 you state that you expect 7% of the women randomized to the control condition to be successful quitters, the use of 7% is very well argued for in the following lines. In the introduction (page 5, lines 39-40) you state that only 3% of the women using the services become smoke free. If I understand correctly, the difference is solely due to the effect of entering a
---

	study, even if you are assigned to the control group. Is that correct? Methods and analysis 6) Page 7, line 54: Seems that a reference has gone missing. 7) Page 9, line 45: I was just curious about the -0.59, representing a state worse than death? And as far as I can see you have not added a reference for the EQ-5D-5L. Also, there is a "(ref)" in line 46. 8) Page 10, line 40: "Costs, QALYs and quit rates per participant". I don't understand what you mean by "quit rate per participant"?
--	---

REVIEWER	Floor van den Brand Maastricht University, The Netherlands
REVIEW RETURNED	07-May-2020

GENERAL COMMENTS	REVIEW bmjopen-2020-038827 A protocol for the economic evaluation of the Smoking Cessation in Pregnancy Incentives Trial (CPIT III) Authors: McMeekin N., Sinclair, L., Bauld, L., Tappin, D., Mitchell, A., Boyd K.A 7 May 2020 The protocol describes an economic evaluation of a RCT on financial incentives to promote smoking cessation in pregnant women. Both a trial-based analysis and a lifetime analysis will be performed. This is an important topic in which little research has been conducted. The protocol of the RCT on which the economic evaluation will be based has also been described in another publication: Sinclair, L., McFadden, M., Tilbrook, H. et al. The smoking cessation in pregnancy incentives trial (CPIT): study protocol for a phase III randomised controlled trial. Trials 21, 183 (2020). https://doi.org/10.1186/s13063-019-4042-8. It is not entirely clear to me why the authors have chosen to publish the economic evaluation as a second protocol instead of including it in the previous research protocol. Introduction  - The Introduction section of the manuscript contains information on the RCT (such as inclusion criteria, power calculation) that I think would better fit in the Methods section. - The authors include 15% loss to follow-up in the calculation, but if an intention-to-treat protocol is used, participants lost to follow-up need to be counted as smokers and not as lost to follow-up. I would also prefer to see the actual power calculation.
--

	Methods and analysis  - The authors could add between which dates the research data will be collected - Page 7 line 42-43: “The economic evaluation will be conducted from the perspective of the UK National Health Service (NHS) and Personal Social Services (PSS), for price year 2020”. The authors could add more information on what the NHS and PSS perspectives consist of. Which types of resource use are relevant for these perspectives? - Line 54 shows an error: “An overview of this analysis is illustrated in Error! Reference source not found.” - Page 8, line 14-15: “Patient level data on resource use and outcomes will be routinely collected during the trial and used in the economic evaluation.” Will the resource use data be continuously measured from the start of the trial until 6 months post-partum? - Page 9, line 9-11: “Unit cost information will be combined with the resource use data collected in the trial to estimate the total cost per participant in each trial arm.” Could the authors be more specific on what resource use is measured and how data are collected? Outcomes  - QALYs are used as a primary outcome for the analysis. However, it is very likely that the 6-month time horizon of the trial is too short to detect changes in quality of life. It would be unfortunate if the authors could not draw conclusions on cost-effectiveness because the EQ-5D-5L is not sensitive enough to detect any changes. Therefore, I would recommend to consider adding an additional non-health related measurement instrument for quality of life that could be more sensitive to detecting improvements in quality of life within the short timeframe of the trial. Analysis of costs and effects  - Page 10 lines 16-21: Could the authors be more specific on their plan for missing data? What sensitivity analyses are planned? - Page 10 lines 23-24: “Missing outcome data will be dealt with using the same methods used in the main trial analysis in consultation with the trial statistician.” Could the authors indicate which methods these are? - Page 11 lines 39-42: “The mother’s portion of the model will include: the costs of smoking cessation services, financial incentives, NRT use and smoking related diseases, plus data on relapse rates and transition to smoking related diseases if the woman relapses”. Which
--	--

	smoking-related diseases are included in the model? Are these the same as in the ESIP model?  - Page 12: the authors describe sensitivity analyses that they potentially will perform. What determines which sensitivity analyses will be performed, or why was this not decided yet?
--	---

VERSION 1 – AUTHOR RESPONSE

Reviewer(s)' Comments to Author:

Reviewer: 1	
Reviewer Name: Mette Rasmussen	
In general:	
1) This is an important research topic evaluating the effects of financial incentives and smoking cessation in a very relevant and vulnerable subgroup of smokers not to mention their unborn children in England, Scotland, and Northern Ireland, using data from a Randomized Controlled Trial (CPIT III). I look forward to reading the results.	Thank you for your kind comments.
2) At first, I was a bit confused about the connection to the CPIT III RCT. I would be nice with a sentence or two in the beginning of the manuscript to explain the connection. I do realize the RCT has been published in a separate protocol, but the most important information regarding the study, including the dates for inclusion and patient consent, should be repeated here.	We have revised the Abstract to make the link between this protocol and CPIT III clearer, and revised the Introduction section in the main text. We have added the dates for recruitment and details of consent, and moved the section explaining the background to CPIT III to the Methods section to make it clearer.
Introduction:	
3) Page 5, lines 22-30: The use of references concerning the prevalence of smokers among pregnant women and for the consequences of smoking during pregnancy is sparse.	Thank you, we have added more references (highlighted in yellow).
4) Page 5, line 45ff: In the evidence section you state that "a recent Cochrane review which identified 33 trials (over 21,600 participants) found that nine trials following 2,273 pregnant smokers confirmed the efficacy of financial incentives in this group (Notley)." Only a small part of the studies in this review does, however, confirm the efficacy of financial incentives. I suggest that you add your rationale for choosing to do this study, in spite of the limited evidence. I don't understand what "(Notley)" in the end of the sentence means, I assume it could be the reference to the Cochrane review?	Thank you, this sentence was not clear; nine studies in the Cochrane review related to pregnant women, not all 33 studies. We have amended this sentence to read more clearly and added in the reference.
5) Page 6, line 19ff: In line 19 you state that you expect 7% of the women randomized to	The 3% quit mentioned earlier on in the manuscript is at 4 weeks (taken from Tappin

the control condition to be successful quitters, the use of 7% is very well argued for in the following lines. In the introduction (page 5, lines 39-40) you state that only 3% of the women using the services become smoke free. If I understand correctly, the difference is solely due to the effect of entering a study, even if you are assigned to the control group. Is that correct?	paper on financial incentives; Tappin DM, MacAskill S, Bauld L, Eadie D, Shipton D, Galbraith L. Smoking prevalence and smoking cessation services for pregnant women in Scotland. Subst Abuse Treat Prev Policy 2010;5:1), and relates to all women who self-report as smokers at maternity booking both those who have considered quitting and those who have not. . Whereas the 7% is from the sample size calculation based on 8.6% quit rate in control arm 34-38 weeks gestation in the CPIT II trial this is a quit rate in late pregnancy. Tim Coleman in 2012 and Michael Ussher in 2015 report similar findings, this later quit rate relates to pregnant smokers who have enrolled in trials of smoking cessation interventions and as such are likely to have considered quitting.
Methods and analysis	
6) Page 7, line 54: Seems that a reference has gone missing.	Thank you, this has now been corrected.
7) Page 9, line 45: I was just curious about the -0.59, representing a state worse than death? And as far as I can see you have not added a reference for the EQ-5D-5L. Also, there is a "(ref)" in line 46.	Thank you I have now added this reference and an explanation of the 'state worse than death'. In terms of a state worse than death it is a 'feature' of the UK value set where members of the general public were asked to value health states using time trade off, where one option was believed to be worse than death. Examples in some studies include: bowel and bladder incontinence, relying on a breathing machine and confused all the time – one interpretation of this result is that people don't have insight into how adaptable they may be in those situations.
8) Page 10, line 40: "Costs, QALYs and quit rates per participant". I don't understand what you mean by "quit rate per participant"?	Thank you, we have revised this sentence to make it clearer; we will present the mean costs, QALYs and quit rate per participants in the trial, dependant on trial arm. Quit rate per participant is the average quit rate per participant in each arm, ie in CPIT II this was 22.5% in the incentives arm and 8.6% in the control arm.

Reviewer #2	
Reviewer Name: Floor van den Brand	
The protocol describes an economic evaluation of a RCT on financial incentives to promote smoking cessation in pregnant women. Both a trial-based analysis and a lifetime analysis will be performed. This is an important	Thank you for your comments.

topic in which little research has been conducted.	
The protocol of the RCT on which the economic evaluation will be based has also been described in another publication: Sinclair, L., McFadden, M., Tilbrook, H. et al. The smoking cessation in pregnancy incentives trial (CPIT): study protocol for a phase III randomised controlled trial. Trials 21, 183 (2020). https://doi.org/10.1186/s13063-019-4042-8. It is not entirely clear to me why the authors have chosen to publish the economic evaluation as a second protocol instead of including it in the previous research protocol.	The use of statistical analysis plans is an accepted practise, and there is a growing trend towards producing health economics analysis plans as well. As the number of health economic analyses in RCTs grow, developing a health economics analysis plan a priori becomes imperative. In CPIT III the word limit for the main study protocol restricted the amount of detail that could be included on the health economics work and we took this opportunity to add to the growing body of published health economics analysis plans, particularly as there will be both a within study and lifetime analysis with data collection from different services.
Introduction	
- The Introduction section of the manuscript contains information on the RCT (such as inclusion criteria, power calculation) that I think would better fit in the Methods section.	Thank you we have moved this to the Methods section.
- The authors include 15% loss to follow-up in the calculation, but if an intention-to-treat protocol is used, participants lost to follow-up need to be counted as smokers and not as lost to follow-up. I would also prefer to see the actual power calculation.	We have added that participants lost to follow-up will be treated as smokers as per the Russell Standard – in line with the main trial protocol. Due to their being some debate in the literature over the use of the Russell Standard as a missing data assumption, the sample size was calculated assuming 15% loss to follow up, which is a conservative approach as if there is no attrition due to the Russell Standard being used, the power will exceed 90%. This approach to sample size calculation was used in the SCIMITAR+ trial published in The Lancet. To demonstrate the merits of this conservative approach, we have added text specifying the power 940 participants would give with 0% attrition.
Methods and analysis	
- The authors could add between which dates the research data will be collected	Thank you for this suggestion, we have added the dates to the Methods section.

- Page 7 line 42-43: “The economic evaluation will be conducted from the perspective of the UK National Health Service (NHS) and Personal Social Services (PSS), for price year 2020”. The authors could add more information on what the NHS and PSS perspectives consist of. Which types of resource use are relevant for these perspectives?	Thank you, we have added more detail; this perspective includes only resources funded by the NHS.
- Line 54 shows an error: “An overview of this analysis is illustrated in Error! Reference source not found.”	Thank you, this has now been corrected.
- Page 8, line 14-15: “Patient level data on resource use and outcomes will be routinely collected during the trial and used in the economic evaluation.” Will the resource use data be continuously measured from the start of the trial until 6 months post-partum?	There are four categories for resource use, and each will cover different time periods as laid out below: Resource data for the cessation services will typically be around 12 weeks’ worth of NRT and cessation support from first contact with cessation services to last contact. Resource use in terms of shopping vouchers will be recorded at each follow-up point, up to and including six-months post-partum. Birth weight and date of delivery will be collected after birth and assumptions on resource use for neo-natal care will be used to attach resource use to any pre-term births. We have made this clear in the manuscript.
- Page 9, line 9-11: “Unit cost information will be combined with the resource use data collected in the trial to estimate the total cost per participant in each trial arm.” Could the authors be more specific on what resource use is measured and how data are collected?	We have revised the earlier ‘Resource use’ section to make this clearer.
Outcomes	

- QALYs are used as a primary outcome for the analysis. However, it is very likely that the 6-month time horizon of the trial is too short to detect changes in quality of life. It would be unfortunate if the authors could not draw conclusions on cost-effectiveness because the EQ-5D-5L is not sensitive enough to detect any changes. Therefore, I would recommend to consider adding an additional nonhealth related measurement instrument for quality of life that could be more sensitive to detecting improvements in quality of life within the short timeframe of the trial.	Thank you for your suggestion, we agree that there is a chance that the EQ-5D-5L will not be sensitive enough to detect any changes in quality of life, however in UK the EQ-5D is the preferred measure of health-related quality of life for estimating QALYs for the National Institute for Health and Care Excellence, and therefore it is best practice to include this in our analysis. We hope that the EQ-5D-5L will detect a change in quality of life for those participants who do quit smoking. Also as recruited is now finished we are unable to add any further outcome measures.
Analysis of costs and effects	
- Page 10 lines 16-21: Could the authors be more specific on their plan for missing data? What sensitivity analyses are planned?	We have added more detail to this section.
- Page 10 lines 23-24: "Missing outcome data will be dealt with using the same methods used in the main trial analysis in consultation with the trial statistician." Could the authors indicate which methods these are?	We have amended this sentence.
- Page 11 lines 39-42: "The mother's portion of the model will include: the costs of smoking cessation services, financial incentives, NRT use and smoking related diseases, plus data on relapse rates and transition to smoking related diseases if the woman relapses". Which smoking-related diseases are included in the model? Are these the same as in the ESIP model?	Yes, we will use the same smoking related diseases as the ESIP model; coronary heart disease, COPD, lung cancer and stroke. These diseases are linked to smoking, and account for 75% of smoking related deaths.

- Page 12: the authors describe sensitivity analyses that they potentially will perform. What determines which sensitivity analyses will be performed, or why was this not decided yet?	Thank you, we agree that this wasn't clear and have made it clearer in the manuscript. The final choice of sensitivity analyses will be determined once we have the trial data – there may be issues that we would like to explore further in the model, such as matters arising as a result of COVID-19 and not being able to take CO monitoring readings.
--	--

VERSION 2 – REVIEW

REVIEWER	Mette Rasmussen Clinical Health Promotion Centre, WHO-CC, The Parker Institute, Bispebjerg and Frederiksberg Hospital, University of Copenhagen, Copenhagen, Denmark Department of Health Science, Clinical Health Promotion Centre, WHO-CC, Region Skåne, Lund University, Lund, Sweden
REVIEW RETURNED	17-Aug-2020

GENERAL COMMENTS	My comments have been adequately addressed. Please fix the two missing references/links (page 6 line 4 and page 7 line 20). Also, I want to inform you that I was not able to read any of the text in figure 2 due to low quality. Good luck with the study, I am looking forward to seeing the results published.
---

REVIEWER	Floor van den Brand Maastricht University, The Netherlands
REVIEW RETURNED	11-Aug-2020

GENERAL COMMENTS	The authors have made good adjustments to the manuscript, making things clearer for the reader. I am also satisfied with the response of the authors to my comments, and I would recommend the manuscript to be accepted for publication.
--